# GANSynth:
# Adversarial Neural Audio Synthesis

**Jesse Engel, Kumar Krishna Agrawal, Shuo Chen, Ishaan Gulrajani, Chris Donahue,
& Adam Roberts**
Google AI
Mountain View, CA 94043, USA

## Abstract

Efficient audio synthesis is an inherently difficult machine learning task, as human perception is sensitive to both global structure and fine-scale waveform coherence. Autoregressive models, such as WaveNet, model local structure but have slow iterative sampling and lack global latent structure. In contrast, Generative Adversarial Networks (GANs) have global latent conditioning and efficient parallel sampling, but struggle to generate locally-coherent audio waveforms. Herein, we demonstrate that GANs can in fact generate high-fidelity and locally-coherent audio by modeling log magnitudes and instantaneous frequencies with sufficient frequency resolution in the spectral domain. Through extensive empirical investigations on the NSynth dataset, we demonstrate that GANs are able to outperform strong WaveNet baselines on automated and human evaluation metrics, and efficiently generate audio several orders of magnitude faster than their autoregressive counterparts.[1]

## 1 Introduction

*Neural audio synthesis*, training generative models to efficiently produce audio with both high-fidelity and global structure, is a challenging open problem as it requires modeling temporal scales over at least five orders of magnitude ($\sim$0.1ms to $\sim$100s). Large advances in the state-of-the art have been pioneered almost exclusively by autoregressive models, such as WaveNet, which solve the scale problem by focusing on the finest scale possible (a single audio sample) and rely upon external conditioning signals for global structure (van den Oord et al., 2016). This comes at the cost of slow sampling speed, since they rely on inefficient ancestral sampling to generate waveforms one audio sample at a time. Due to their high quality, a lot of research has gone into speeding up generation, but the methods introduce significant overhead such as training a secondary student network or writing highly customized low-level kernels (van den Oord et al., 2018; Paine et al., 2016). Furthermore, since these large models operate at a fine timescale, their autoencoder variants are restricted to only modeling local latent structure due to memory constraints (Engel et al., 2017).

On the other end of the spectrum, Generative Adversarial Networks (GANs) (Goodfellow et al., 2014) have seen great recent success at generating high resolution images (Radford et al., 2016; Arjovsky et al., 2017; Gulrajani et al., 2017; Berthelot et al., 2017; Kodali et al., 2017; Karras et al., 2018a; Miyato et al., 2018). Typical GANs achieve both efficient parallel sampling and global latent control by conditioning a stack of transposed convolutions on a latent vector, The potential for audio GANs extends further, as adversarial costs have unlocked intriguing domain transformations for images that could possibly have analogues in audio (Isola et al., 2017; Zhu et al., 2017; Wolf et al., 2017; Jin et al., 2017). However, attempts to adapt image GAN architectures to generate waveforms in a straightforward manner (Donahue et al., 2019) fail to reach the same level of perceptual fidelity as their image counterparts.

---

[1]Online resources:
Colab Notebook: `http://goo.gl/magenta/gansynth-demo`,
Audio Examples: `http://goo.gl/magenta/gansynth-examples`,
Code: `http://goo.gl/magenta/gansynth-code`

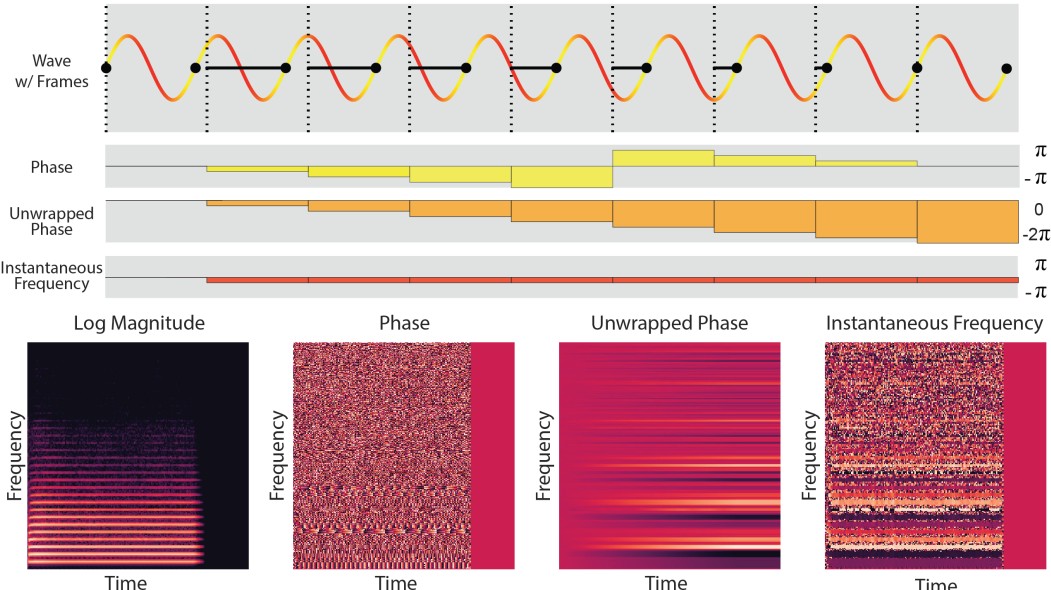

Figure 1: Frame-based estimation of audio waveforms. Much of sound is made up of locally-coherent waves with a local periodicity, pictured as the red-yellow sinusoid with black dots at the start of each cycle. Frame-based techniques, whether they be transposed convolutions or STFTs, have a given frame size and stride, here depicted as equal with boundaries at the dotted lines. The alignment between the two (phase, indicated by the solid black line and yellow boxes), precesses in time since the periodicity of the audio and the output stride are not exactly the same. Transposed convolutional filters thus have the difficult task of covering all the necessary frequencies *and* all possible phase alignments to preserve phase coherence. For an STFT, we can unwrap the phase over the $2\pi$ boundary (orange boxes) and take its derivative to get the instantaneous radial frequency (red boxes), which expresses the constant relationship between audio frequency and frame frequency. The spectra are shown for an example trumpet note from the NSynth dataset.

## 1.1 GENERATING INSTRUMENT TIMBRES

GAN researchers have made rapid progress in image modeling by evaluating models on focused datasets with limited degrees of freedom, and gradually stepping up to less constrained domains. For example, the popular CelebA dataset (Liu et al., 2015) is restricted to faces that have been centered and cropped, removing variance in posture and pose, and providing a common reference for qualitative improvements (Radford et al., 2016; Karras et al., 2018a) in generating realistic texture and fine-scale features. Later models then built on that foundation to generalize to broader domains (Karras et al., 2018b; Brock et al., 2019).

The NSynth dataset (Engel et al., 2017)[2] was introduced with similar motivation for audio. Rather than containing all types of audio, NSynth consists solely of individual notes from musical instruments across a range of pitches, timbres, and volumes. Similar to CelebA, all the data is aligned and cropped to reduce variance and focus on fine-scale details, which in audio corresponds to timbre and fidelity. Further, each note is also accompanied by an array of attribute labels to enable exploring conditional generation.

The original NSynth paper introduced both autoregressive WaveNet autoencoders and bottleneck spectrogram autoencoders, but without the ability to unconditionally sample from a prior. Follow up work has explored diverse approaches including frame-based regression models (Defossez et al., 2018), inverse scattering networks (Andreux & Mallat, 2018), VAEs with perceptual priors (Esling et al., 2018), and adversarial regularization for domain transfer (Mor et al., 2019). This work builds on these efforts by introducing adversarial training and exploring effective representations for non-causal convolutional generation as typical found in GANs.

---

[2]https://magenta.tensorflow.org/datasets/nsynth

## 1.2 EFFECTIVE AUDIO REPRESENTATIONS FOR GANS

Unlike images, most audio waveforms—such as speech and music—are highly periodic. Convolutional filters trained for different tasks on this data commonly learn to form logarithmically-scaled frequency selective filter banks spanning the range of human hearing (Dieleman & Schrauwen, 2014; Zhu et al., 2016). Human perception is also highly sensitive to discontinuities and irregularities in periodic waveforms, so maintaining the regularity of periodic signals over short to intermediate timescales (1ms - 100ms) is crucial. Figure 1 shows that when the stride of the frames does not exactly equal a waveform's periodicity, the alignment (phase) of the two precesses over time. This condition is assured as at any time there are typically many different frequencies in a given signal. This is a challenge for a synthesis network, as it must learn all the appropriate frequency and phase combinations and activate them in just the right combination to produce a coherent waveform. This phase precession is exactly the same phenomena observed with a short-time Fourier transform (STFT), which is composed of strided filterbanks just like convolutional networks. Phase precession also occurs in situations where filterbanks overlap (window or kernel size < stride).

In the middle of Figure 1, we diagram another approach to generating coherent waveforms loosely inspired by the phase vocoder (Dolson, 1986). A pure tone produces a phase that precesses. Unwrapping the phase, by adding $2\pi$ whenever it crosses a phase discontinuity, causes the precessing phase to grow linearly. We then observe that the derivative of the unwrapped phase with respect to time remains constant and is equal to the angular difference between the frame stride and signal periodicity. This is commonly referred to as the *instantaneous angular frequency*, and is a time varying measure of the true signal oscillation. With a slight abuse of terminology we will simply refer to it as the instantaneous frequency (IF) (Boashash, 1992). Note that for the spectra at the bottom of Figure 1, the pure harmonic frequencies of a trumpet cause the wrapped phase spectra to oscillate at different rates while the unwrapped phase smoothly diverges and the IF spectra forms solid bands where the harmonic frequencies are present.

## 1.3 CONTRIBUTIONS

In this paper, we investigate the interplay of architecture and representation in synthesizing coherent audio with GANs. Our key findings include:

- Generating log-magnitude spectrograms and phases directly with GANs can produce more coherent waveforms than directly generating waveforms with strided convolutions.

- Estimating IF spectra leads to more coherent audio still than estimating phase.

- It is important to keep harmonics from overlapping. Both increasing the STFT frame size and switching to mel frequency scale improve performance by creating more separation between the lower harmonic frequencies. Harmonic frequencies are multiples of the fundamental, so low pitches have tightly-spaced harmonics, which can cause blurring and overlap.

- On the NSynth dataset, GANs can outperform a strong WaveNet baseline in automatic and human evaluations, and generate examples ∼54,000 times faster.

- Global conditioning on latent and pitch vectors allow GANs to generate perceptually smooth interpolation in timbre, and consistent timbral identity across pitch.

## 2 EXPERIMENTAL DETAILS

### 2.1 DATASET

We focus our study on the NSynth dataset, which contains 300,000 musical notes from 1,000 different instruments aligned and recorded in isolation. NSynth is a difficult dataset composed of highly diverse timbres and pitches, but it is also highly structured with labels for pitch, velocity, instrument, and acoustic qualities (Liu et al., 2015; Engel et al., 2017). Each sample is four seconds long, and sampled at 16kHz, giving 64,000 dimensions. As we wanted to included human evaluations on audio quality, we restricted ourselves to training on the subset of acoustic instruments and fundamental pitches ranging from MIDI 24-84 (∼32-1000Hz), as those timbres are most likely to sound natural

to an average listener. This left us with 70,379 examples from instruments that are mostly strings, brass, woodwinds, and mallets. We created a new test/train 80/20 split from shuffled data, as the original split was divided along instrument type, which isn't desirable for this task.

## 2.2 ARCHITECTURE AND REPRESENTATIONS

Taking inspiration from successes in image generation, we adapt the progressive training methods of Karras et al. (2018a) to instead generate audio spectra [3]. While e search over a variety of hyper-parameter configurations and learning rates, we direct readers to the original paper for an in-depth analysis (Karras et al., 2018a), and the appendix for complete details.

Briefly, the model samples a random vector $\mathbf{z}$ from a spherical Gaussian, and runs it through a stack of transposed convolutions to upsample and generate output data $x = G(\mathbf{z})$, which is fed into a discriminator network of downsampling convolutions (whose architecture mirrors the generator's) to estimate a divergence measure between the real and generated distributions (Arjovsky et al., 2017). As in Karras et al. (2018a), we use a gradient penalty (Gulrajani et al., 2017) to promote Lipschitz continuity, and pixel normalization at each layer. We also try training both progressive and non-progressive variants, and see comparable quality in both. While it is not essential for success, we do see slightly better convergence time and sample diversity for progressive training, so for the remainder of the paper, all models are compared with progressive training.

Unlike Progressive GAN, our method involves conditioning on an additional source of information. Specifically, we append a one-hot representation of musical pitch to the latent vector, with the musically-desirable goal of achieving independent control of pitch and timbre. To encourage the generator to use the pitch information, we also add an auxiliary classification (Odena et al., 2017) loss to the discriminator that tries to predict the pitch label.

For spectral representations, we compute STFT magnitudes and phase angles using TensorFlow's built-in implementation. We use an STFT with 256 stride and 1024 frame size, resulting in 75% frame overlap and 513 frequency bins. We trim the Nyquist frequency and pad in time to get an "image" of size (256, 512, 2). The two channel dimension correspond to magnitude and phase. We take the log of the magnitude to better constrain the range and then scale the magnitudes to be between -1 and 1 to match the tanh output nonlinearity of the generator network. The phase angle is also scaled to between -1 and 1 and we refer to these variants as "**phase**" models. We optionally unwrap the phase angle and take the finite difference as in Figure 1; we call the resulting models "instantaneous frequency" ("**IF**") models. We also find performance is sensitive to having sufficient frequency resolution at the lower frequency range. Maintaining 75% overlap we are able to double the STFT frame size and stride, resulting in spectral images with size (128, 1024, 2), which we refer to as high frequency resolution, "**+ H**", variants. Lastly, to provide even more separation of lower frequencies we transform both the log magnitudes and instantaneous frequencies to a mel frequency scale without dimensional compression (1024 bins), which we refer to as "**IF-Mel**" variants. To convert back to linear STFTs we just use the approximate inverse linear transformation, which, perhaps surprisingly does not harm audio quality significantly.

It is important for us to compare against strong baselines, so we adapt WaveGAN (Donahue et al., 2019), the current state of the art in waveform generation with GANs, to accept pitch conditioning and retrain it on our subset of the NSynth dataset. We also independently train our own waveform generating GANs off the progressive codebase and our best models achieve similar performance to WaveGAN without progressive training, so we opt to primarily show numbers from WaveGAN instead (see appendix Table 2 for more details).

Beyond GANs, WaveNet (van den Oord et al., 2016) is currently the state of the art in generative modeling of audio. Prior work on the NSynth dataset used an WaveNet autoencoder to interpolate between sounds (Engel et al., 2017), but is not a generative model as it requires conditioning on the original audio. Thus, we create strong WaveNet baselines by adapting the architecture to accept the same one-hot pitch conditioning signal as the GANs. We train variants using both a categorical 8-bit mu law and 16-bit mixture of logistics for the output distributions, but find that the 8-bit model is more stable and outperforms the 16-bit model (see appendix Table 2 for more details).

---

[3]Code: http://goo.gl/magenta/gansynth-code

## 3    METRICS

Evaluating generative models is itself a difficult problem: because our goals (perceptually-realistic audio generation) are hard to formalize, the most common evaluation metrics tend to be heuristic and have "blind spots" (Theis et al., 2016). To mitigate this, we evaluate all of our models against a diverse set of metrics, each of which captures a distinct aspect of model performance. Our evaluation metrics are as follows:

- **Human Evaluation**    We use human evaluators as our gold standard of audio quality because it is notoriously hard to measure in an automated manner. In the end, we are interested in training networks to synthesize coherent waveforms, specifically because human perception is extremely sensitive to phase irregularities and these irregularities are disruptive to a listener. We used Amazon Mechanical Turk to perform a comparison test on examples from all models presented in Table 1 (this includes the hold-out dataset). The participants were presented with two 4s examples corresponding to the same pitch. On a five-level Likert scale, the participants evaluate the statement "*Sample A has better audio quality / has less audio distortions than Sample B*". For the study, we collected 3600 ratings and each model is involved in 800 comparisons.

- **Number of Statistically-Different Bins (NDB)**    We adopt the metric proposed by Richardson & Weiss (2018) to measure the diversity of generated examples: the training examples are clustered into $k = 50$ *Voronoi cells* by $k$-means in log-spectrogram space, the generated examples are also mapped into the same space and are assigned to the nearest cell. NDB is reported as the number of cells where the number of training examples is statistically significantly different from the number of generated examples by a two-sample Binomial test.

- **Inception Score (IS)**    (Salimans et al., 2016) propose a metric for evaluating GANs which has become a de-facto standard in GAN literature (Gulrajani et al., 2017; Miyato et al., 2018; Karras et al., 2018a). Generated examples are run through a pretrained Inception classifier and the Inception Score is defined as the mean KL divergence between the image-conditional output class probabilities and the marginal distribution of the same. IS penalizes models whose examples aren't each easily classified into a single class, as well as models whose examples collectively belong to only a few of the possible classes. Though we still call our metric "IS" for consistency, we replace the Inception features with features from a pitch classifier trained on spectrograms of our acoustic NSynth dataset.

- **Pitch Accuracy (PA) and Pitch Entropy (PE)**    Because the Inception Score can conflate models which don't produce distinct pitches and models which produce only a few pitches, we also separately measure the accuracy of the same pretrained pitch classifier on generated examples (PA) and the entropy of its output distribution (PE).

- **Fréchet Inception Distance (FID)** (Heusel et al., 2017) propose a metric for evaluating GANs based on the 2-Wasserstein (or Fréchet) distance between multivariate Gaussians fit to features extracted from a pretrained Inception classifier and show that this metric correlates with perceptual quality and diversity on synthetic distributions. As with Inception Score, we use pitch-classifier features instead of Inception features.

## 4    RESULTS

Table 1 presents a summary of our results on all model and representation variants. Our most discerning measure of audio quality, human evaluation, shows a clear trend, summarized in Figure 2. Quality decreases as output representations move from IF-Mel, IF, Phase, to Waveform. The highest quality model, IF-Mel, was judged comparably but slightly inferior to real data. The WaveNet baseline produces high-fidelity sounds, but occasionally breaks down into feedback and self oscillation, resulting in a score that is comparable to the IF GANs.

While there is no a priori reason that sample diversity should correlate with audio quality, we indeed find that NDB follows the same trend as the human evaluation. Additionally, high frequency resolution improves the NDB score across models types. The WaveNet baseline receives the worst NDB score. Even though the generative model assigns high likelihood to all the training data, the

Table 1: Metrics for different models. "+ H" stands for higher frequency resolution, and "Real Data" is drawn from the test set.

| Examples | Human Eval (wins) | NDB | FID | IS | PA | PE |
|---|---|---|---|---|---|---|
| Real Data | 549 | 2.2 | 13 | 47.1 | 98.2 | 0.22 |
| **IF-Mel + H** | **485** | **29.3** | 167 | 38.1 | 97.9 | 0.40 |
| IF + H | 308 | 36. 0 | **104** | **41.6** | **98.3** | **0.32** |
| Phase + H | 225 | 37.6 | 592 | 36.2 | 97.6 | 0.44 |
| IF-Mel | 479 | 37.0 | 600 | 29.6 | 94.1 | 0.63 |
| IF | 283 | 37.0 | 708 | 36.3 | 96.8 | 0.44 |
| Phase | 203 | 41.4 | 687 | 24.4 | 94.4 | 0.77 |
| WaveNet | 359 | 45.9 | 320 | 29.1 | 92.7 | 0.70 |
| WaveGAN | 216 | 43.0 | 461 | 13.7 | 82.7 | 1.40 |

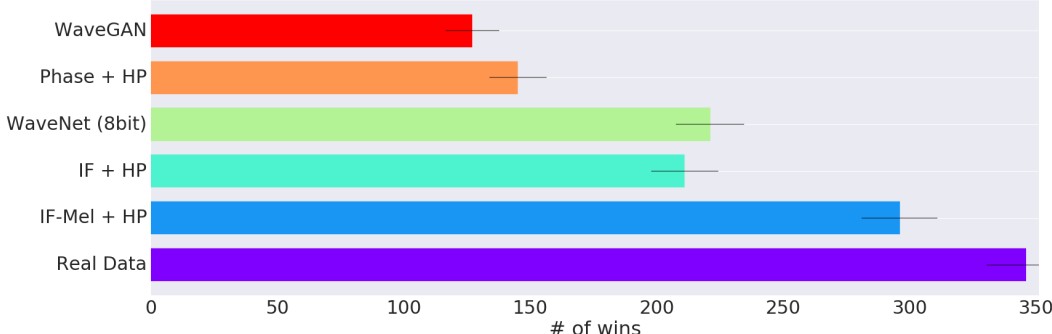

Figure 2: Number of wins on pair-wise comparison across different output representations and baselines. Ablation comparing highest performing models of each type. Higher scores represent better perceptual quality to participants. The ranking observed here correlates well with the evaluation on quantitative metrics as in Table 1.

autoregressive sampling itself has a tendency gravitate to the same type of oscillation for each given pitch conditioning, leading to an extreme lack of diversity. Histograms of the sample distributions showing peaky distributions for the different models can be found in the appendix.

FID provides a similar story to the first two metrics with significantly lower scores for for IF models with high frequency resolution. Comparatively, Mel scaling has much less of an effect on the FID then it does in the listener study. Phase models have high FID, even at high frequency resolution, reflecting their poor sample quality.

Many of the models do quite well on the classifier metrics of IS, Pitch Accuracy, and Pitch Entropy, because they have explicit conditioning telling them what pitches to generate. All of the high-resolution models actually generate examples classified with similar accuracy to the real data. As this accuracy and entropy can be a strong function of the distribution of generated examples, which most certainly does not match the training distribution due to mode collapse and other issues, there is little discriminative information to gain about sample quality from differences among such high scores. The metrics do provide a rough measure of which models are less reliably generating classifiable pitches, which seems to be the low frequency models to some extent and the baselines.

## 5 QUALITATIVE ANALYSIS

While we do our best to visualize qualitative audio concepts, **we highly recommend the reader to listen to the accompanying audio examples** provided at `https://goo.gl/magenta/gansynth-examples`.

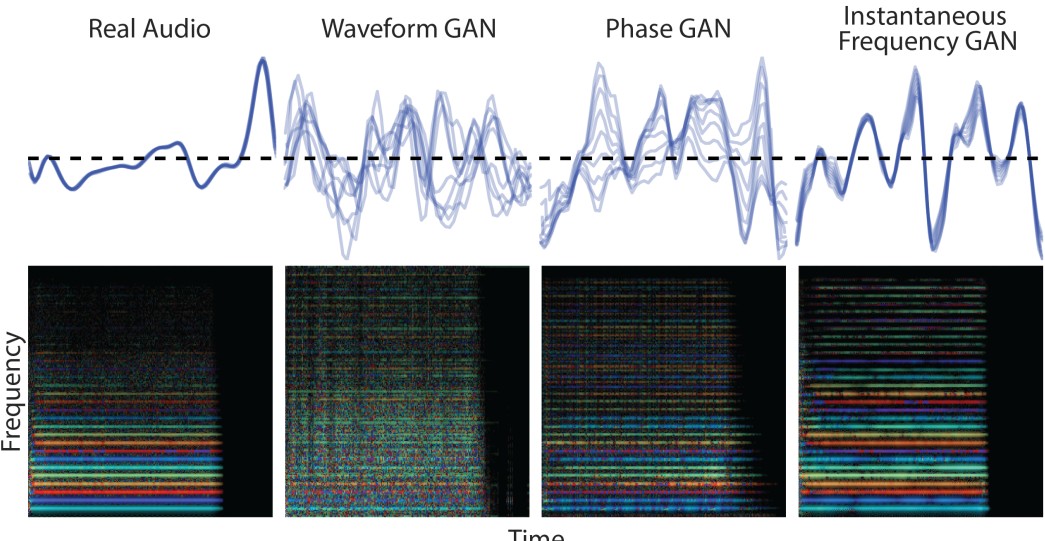

Figure 3: Phase coherence. Examples are selected to be roughly similar between the models for illustrative purposes. The top row shows the waveform modulo the fundamental periodicity of the note (MIDI C60), for 1028 examples taken in the middle of the note. Notice that the real data completely overlaps itself as the waveform is extremely periodic. The WaveGAN and PhaseGAN, however, have many phase irregularities, creating a blurry web of lines. The IFGAN is much more coherent, having only small variations from cycle-to-cycle. In the Rainbowgrams below, the real data and IF models have coherent waveforms that result in strong consistent colors for each harmonic, while the PhaseGAN has many speckles due to phase discontinuities, and the WaveGAN model is quite irregular.

## 5.1 PHASE COHERENCE

Figure 3 visualizes the phase coherence of examples from different GAN variants. It is clear from the waveforms at the top, which are wrapped at the fundamental frequency, that the real data and IF models produce waveforms that are consistent from cycle-to-cycle. The PhaseGAN has some phase discontinuities, while the WaveGAN is quite irregular. Below we use Rainbowgrams (Engel et al., 2017) to depict the log magnitude of the frequencies as brightness and the IF as the color on a rainbow color map. This visualization helps to see clear phase coherence of the harmonics in the real data and IFGAN by the strong consistent colors. In contrast, the PhaseGAN discontinuities appear as speckled noise, and the WaveGAN appears largely incoherent.

## 5.2 INTERPOLATION

As discussed in the introduction, GANs also allow conditioning on the same latent vector the entire sequence, as opposed to only short subsequences for memory intensive autoregressive models like WaveNet. WaveNet autoencoders, such as ones in (Engel et al., 2017), learn local latent codes that control generation on the scale of milliseconds but have limited scope, and have a structure of their own that must be modelled and does not fit a compact prior. In Figure 4 we take a pretrained WaveNet autoencoder [5] and compare interpolating between examples in the raw waveform (top), the distributed latent code of a WaveNet autoencoder, and the global code of an IF-Mel GAN.

Interpolating the waveform is perceptually equivalent to mixing between the amplitudes of two distinct sounds. WaveNet improves upon this for the two notes by mixing in the space of timbre, but the linear interpolation does not correspond to the complex prior on latents, and the intermediate sounds have a tendency to fall apart, oscillate and whistle, which are the natural failure modes for a WaveNet model. However, the GAN model has a spherical gaussian prior which is decoded to

---

[5]https://github.com/tensorflow/magenta/tree/master/magenta/models/nsynth

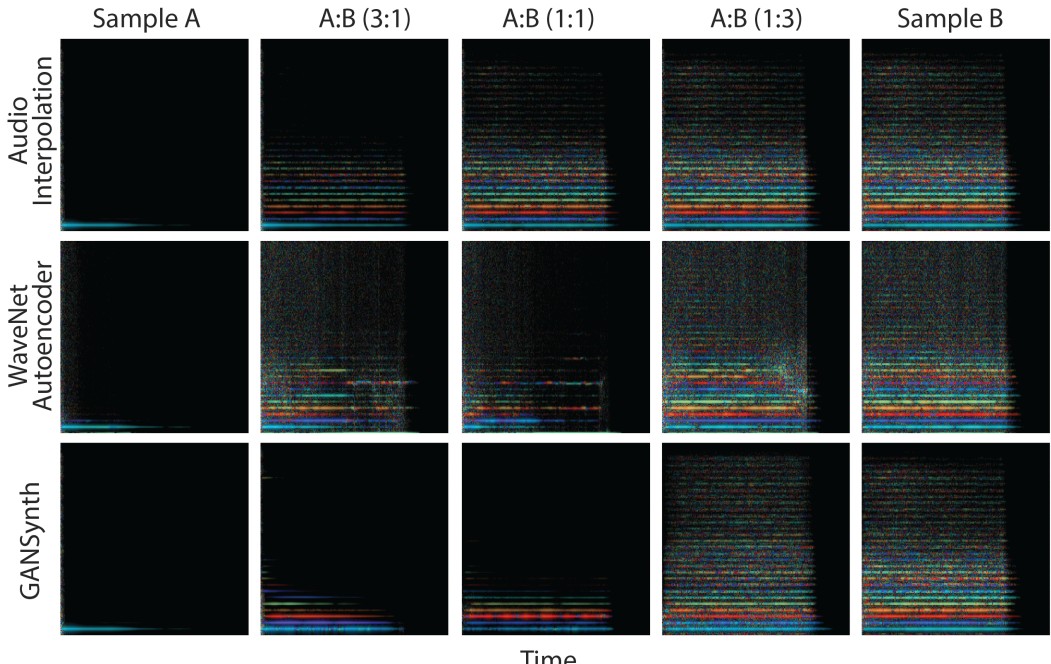

Figure 4: Global interpolation. Examples available for listening[4]. Interpolating between waveforms perceptually results in crossfading the volumes of two distinct sounds (rainbowgrams at top). The WaveNet autoencoder (middle) only has local conditioning distributed in time, and no compact prior over those time series, so linear interpolation ventures off the true prior / data manifold, and produces in-between sounds that are less realistic examples and feature the default failure mode of autoregressive wavenets (feedback harmonics). Meanwhile, the IF-Mel GAN (bottom) has global conditioning so interpolating in perceptual attributes while staying along the prior at all intermediate points, so they produce high-fidelity audio examples like the endpoints.

produce the entire sound, and spherical interpolation stays well-aligned with the prior. Thus, the perceptual change during interpolation is smooth and all intermediate latent vectors are decoded to produce sounds without additional artifacts. As a more musical example, in the audio examples, we interpolate the timbre between 15 random points in latent space while using the pitches from the prelude to Bach's Suite No. 1 in G major [6]. As seen in appendix Figure 7, the timbre of the sounds morph smoothly between many instruments while the pitches consistently follow the composed piece.

## 5.3 CONSISTENT TIMBRE ACROSS PITCH

While timbre slightly varies for a natural instrument across register, on the whole it remains consistent, giving the instrument its unique character. In the audio examples [7], we fix the latent conditioning variable and generate examples by varying the pitch conditioning over five octaves. It's clear that the timbral identity of the GAN remains largely intact, creating a unique instrument identity for the given point in latent space. As seen in appendix Figure 7, the Bach prelude rendered with a single latent vector has a consistent harmonic structure across a range of pitches.

## 6 FAST GENERATION

One of the advantages of GANs with upsampling convolutions over autoregressive models is that the both the training and generation can be processed in parallel for the entire audio sample. This

---

[6] http://www.jsbach.net/midi/midi_solo_cello.html
[7] https://goo.gl/magenta/gansynth-examples

is quite amenable to modern GPU hardware which is often I/O bound with iterative autoregressive algorithms. This can be seen when we synthesize a single four second audio sample on a TitanX GPU and the latency to completion drops from 1077.53 seconds for the WaveNet baseline to 20 milliseconds for the IF-Mel GAN making it around 53,880 times faster. Previous applications of WaveNet autoencoders trained on the NSynth dataset for music performance relied on prerendering all possible sounds for playback due to the long synthesis latency [8]. This work opens up the intriguing possibility for realtime neural network audio synthesis on device, allowing users to explore a much broader pallete of expressive sounds.

## 7    RELATED WORK

Much of the work on deep generative models for audio tends to focus on speech synthesis (van den Oord et al., 2018; Sotelo et al., 2017; Wang et al., 2017). These datasets require handling variable length conditioning (phonemes/text) and audio, and often rely on recurrent and/or autoregressive models for variable length inputs and outputs. It would be interesting to compare adversarial audio synthesis to these methods, but we leave this to future work as adapting GANs to use variable-length conditioning or recurrent generators is a non-trivial extension of the current work.

In comparison to speech, audio generation for music is relatively under-explored. van den Oord et al. (2016) and Mehri et al. (2017) propose autoregressive models and demonstrate their ability to synthesize musical instrument sounds, but these suffer from the aforementioned slow generation. Donahue et al. (2019) first applied GANs to audio generation with coherent results, but fell short of the audio fidelity of autoregressive likelihood models.

Our work also builds on multiple recent advances in GAN literature. Gulrajani et al. (2017) propose a modification to the loss function of GANs and demonstrate improved training stability and architectural robustness. Karras et al. (2018a) further introduce *progressive training*, in which successive layers of the generator and discriminator are learned in a curriculum, leading to improved generation quality given a limited training time. They also propose a number of architectural tricks to further improve quality, which we employ in our best models.

The NSynth dataset was first introduced as a "CelebA of audio" (Liu et al., 2015; Engel et al., 2017), and used WaveNet autoencoders to interpolate between timbres of musical instruments, but with very slow sampling speeds. Mor et al. (2019) expanded on this work by incoporating an adversarial domain confusion loss to achieve timbre transformations between a wide range of audio sources. Defossez et al. (2018) achieve significant sampling speedups ($\sim$2,500x) over wavenet autoencoders by training a frame-based regression model to map from pitch and instrument labels to raw waveforms. They consider a unimodal likelihood regression loss in log spectrograms and back-propagate through the STFT, which yeilds good frequency estimation, but provides no incentive to learn phase coherency or handle multimodal distributions. Their architecture also requires a large amount of channels, slowing down sample generation and training.

## 8    CONCLUSION

By carefully controlling the audio representation used for generative modeling, we have demonstrated high-quality audio generation with GANs on the NSynth dataset, exceeding the fidelity of a strong WaveNet baseline while generating samples tens of thousands of times faster. While this is a major advance for audio generation with GANs, this study focused on a specific controlled dataset, and further work is needed to validate and expand it to a broader class of signals including speech and other types of natural sound. This work also opens up possible avenues for domain transfer and other exciting applications of adversarial losses to audio. Issues of mode collapse and diversity common to GANs exist for audio as well, and we leave it to further work to consider combining adversarial losses with encoders or more straightforward regression losses to better capture the full data distribution.

---

[8] http://g.co/nsynthsuper

ACKNOWLEDGMENTS

We would like to thank Rif A. Saurous and David Berthelot for fruitful discussions and help in reviewing the manuscript.

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

## A  MEASURING DIVERSITY ACROSS GENERATED EXAMPLES

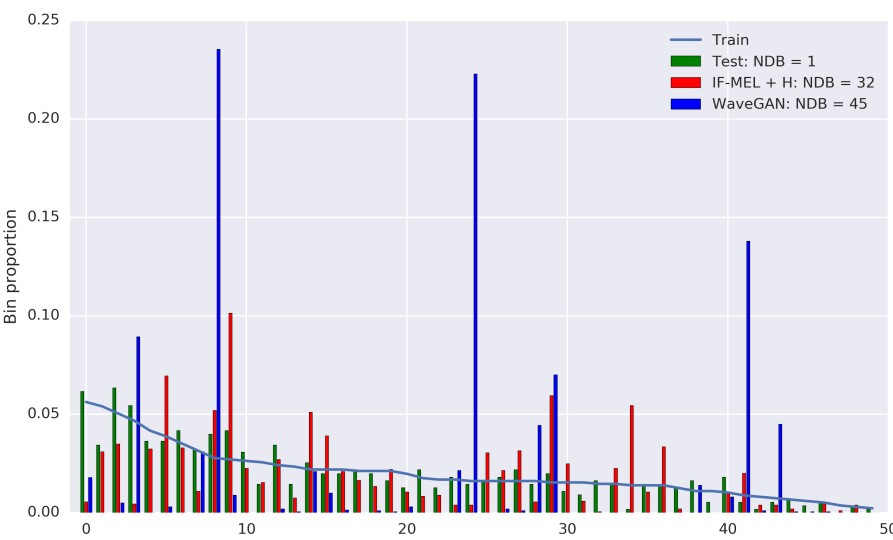

Figure 5: NDB bin proportions for the IF-Mel + H model and the WaveGAN baseline (evaluated with examples of pitch 60).

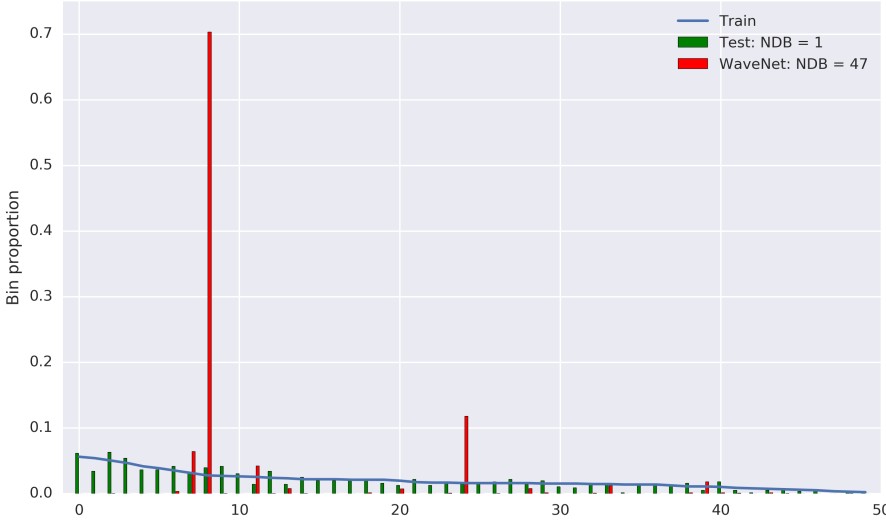

Figure 6: NDB bin proportions for the WaveNet baseline (evaluated with examples of pitch 60).

## B  TIMBRAL SIMILARITY ACROSS PITCH

Bach Prelude -- Single Latent Vector

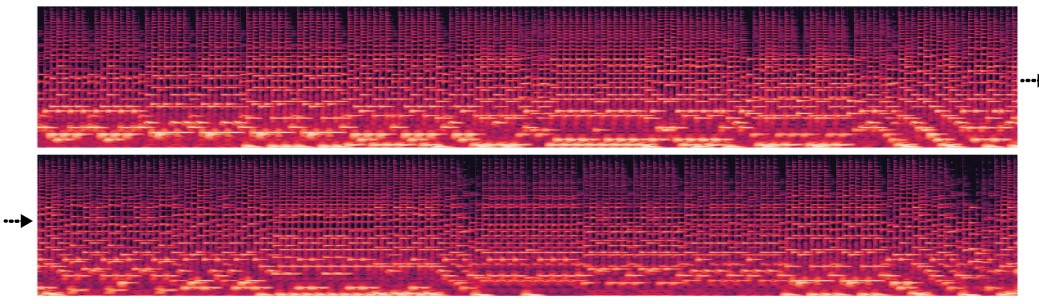

Bach Prelude -- Latent Interpolation

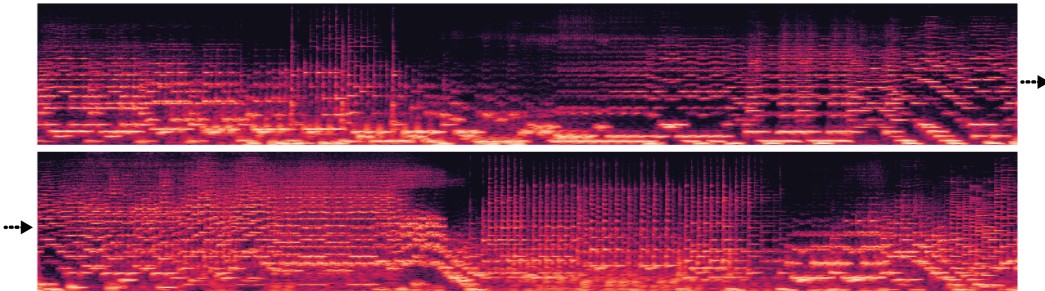

Figure 7: The first 20 seconds (10 seconds per a row) of the prelude to Bach's Suite No. 1 in G major [9], for pitches synthesized with a single latent vector (top), and with spherical interpolation in latent space (bottom). The timbre is constant for a single latent vector, shown by the consistency of the upper harmonic structure, while it varies dramatically as the latent vector changes. Listening examples are provided at `https://goo.gl/magenta/gansynth-examples`

## C  BASELINE MODEL COMPARISONS

Table 2: Comparison of models generating waveforms directly. Our Waveform GAN baseline performs similar to the WaveGAN baseline, but the progressive training does not improve performance, so we only compare to the WaveGAN baseline for the paper. The 8-bit categorical WaveNet outperforms the 16-bit mixture of logistics, likely due to the decreased stability of the 16-bit model with only pitch conditioning, despite the increased fidelity.

| Examples | NDB | FID | IS | PA | PE |
|---|---|---|---|---|---|
| WaveGAN | 43.0 | 461 | 13.7 | 82.7 | 1.40 |
| Waveform NoProg | 48.2 | 447 | 14.8 | 96.3 | 1.61 |
| Waveform Prog | 45.0 | 375 | 2.5 | 56.7 | 3.59 |
| WaveNet 8-bit | 44.8 | 320 | 29.1 | 92.7 | 0.70 |
| WaveNet 16-bit | 45.9 | 656 | 9.5 | 64.6 | 1.71 |

## D   TRAINING DETAILS

GAN architectures were directly adapted from an open source implementation in Tensorflow [10]. Full details are given in Table 3, including adding a pitch classifier to the end of the discriminator as in AC-GAN. All models were trained with the ADAM optimizer (Kingma & Ba, 2014). We sweep over learning rates (2e-4, 4e-4, 8e-4) and weights of the auxiliary classifier loss (0.1, 1.0, 10), and find that for all variants (spectral representation, progressive/no progressive, frequency resolution) a learning rate of 8e-4 and classifier loss of 10 perform the best.

As in the original progressive GAN paper, both networks use box upscaling/downscaling and the generators use pixel normalization,

$$x = x_{nhwc}/(\frac{1}{C} \sum_c x^2_{nhwc})^{0.5} \tag{1}$$

where $n$, $h$, $w$, and $c$ refer to the batch, height, width, and channel dimensions respectively, $x$ is the activations, and $C$ is the total number of channels. The discriminator also appends the standard deviation of the minibatch activations as a scalar channel near the end of the convolutional stack as seen in Table 3.

Since we find it helpful to use a Tanh output nonlinearity for the generator, we normalize real data before passing to the discriminator. We measure the maximum range over 100 examples and independently shift and scale the log-magnitudes and phases to [-0.8, 0.8] to allow for outliers and use more of the linear regime of the Tanh nonlinearity.

We train each GAN variant for 4.5 days on a single V100 GPU, with a batch size of 8. For non-progressive models, this equates to training on ∼5M examples. For progressive models, we train on 1.6M examples per a stage (7 stages), 800k during alpha blending and 800k after blending. At the last stage we continue training until the 4.5 days completes. Because the earlier stages train faster, the progressive models train on ∼11M examples.

For the WaveNet baseline, we also adapt the open source Tensorflow implementation [11]. The decoder is composed of 30 layers of dilated convolution, each of 512 channels and receptive field of 3, and each with a 1x1 convolution skip connection to the output. The layers are divided into 3 stacks of 10, with dilation in each stack increasing from $2^0$ to $2^9$, and then repeating. We replace the audio encoder stack with a conditioning stack operating on a one-hot pitch conditioning signal distributed in time (3 seconds on, 1 second off). The conditioning stack is 5 layers of dilated convolution, increasing to $2^5$, and then 3 layers of regular convolution, all with 512 channels. This conditioning signal is then passed through a 1x1 convolution for each layer of the decoder and added to the output of each layer, as in other implementations of WaveNet conditioning. For the 8-bit model we use mu-law encoding of the audio and a categorical loss, while for the 16-bit model we use a quantized mixture of 10 logistics (Salimans et al., 2017). WaveNets converged to 150k iterations in 2 days with 32 V100 GPUs trained with synchronous SGD with batch size 1 per GPU, for a total batch size of 32.

---

[10]https://github.com/tensorflow/models/tree/master/research/gan/progressive_gan

[11]https://github.com/tensorflow/magenta/tree/master/magenta/models/nsynth

Table 3: Model architecture for hi-frequency resolution. Low frequency resolution starts with a width of 4, and height of 8, but is otherwise the same. "PN" stands for pixel norm, and "LReLU" stands for leaky rectified linear unit, with a slope of 0.2. The latent vector Z has 256 dimensions and the pitch conditioning is a 61 dimensional one-hot vector.

| Generator | Output Size | $k_{Width}$ | $k_{Height}$ | $k_{Filters}$ | Nonlinearity |
|---|---|---|---|---|---|
| concat(Z, Pitch) | (1, 1, 317) | - | - | - | - |
| conv2d | (2, 16, 256) | 2 | 16 | 256 | PN(LReLU) |
| conv2d | (2, 16, 256) | 3 | 3 | 256 | PN(LReLU) |
| upsample 2x2 | (4, 32, 256) | - | - | - | - |
| conv2d | (4, 32, 256) | 3 | 3 | 256 | PN(LReLU) |
| conv2d | (4, 32, 256) | 3 | 3 | 256 | PN(LReLU) |
| upsample 2x2 | (8, 64, 256) | - | - | - | - |
| conv2d | (8, 64, 256) | 3 | 3 | 256 | PN(LReLU) |
| conv2d | (8, 64, 256) | 3 | 3 | 256 | PN(LReLU) |
| upsample 2x2 | (16, 128, 256) | - | - | - | - |
| conv2d | (16, 128, 256) | 3 | 3 | 256 | PN(LReLU) |
| conv2d | (16, 128, 256) | 3 | 3 | 256 | PN(LReLU) |
| upsample 2x2 | (32, 256, 256) | - | - | - | - |
| conv2d | (32, 256, 128) | 3 | 3 | 128 | PN(LReLU) |
| conv2d | (32, 256, 128) | 3 | 3 | 128 | PN(LReLU) |
| upsample 2x2 | (64, 512, 128) | - | - | - | - |
| conv2d | (64, 512, 64) | 3 | 3 | 64 | PN(LReLU) |
| conv2d | (64, 512, 64) | 3 | 3 | 64 | PN(LReLU) |
| upsample 2x2 | (128, 1024, 64) | - | - | - | - |
| conv2d | (128, 1024, 32) | 3 | 3 | 32 | PN(LReLU) |
| conv2d | (128, 1024, 32) | 3 | 3 | 32 | PN(LReLU) |
| generator output | (128, 1024, 2) | 1 | 1 | 2 | Tanh |
| **Discriminator** | | | | | |
| image | (128, 1024, 2) | - | - | - | - |
| conv2d | (128, 1024, 32) | 1 | 1 | 32 | - |
| conv2d | (128, 1024, 32) | 3 | 3 | 32 | LReLU |
| conv2d | (128, 1024, 32) | 3 | 3 | 32 | LReLU |
| downsample 2x2 | (64, 512, 32) | - | - | - | - |
| conv2d | (64, 512, 64) | 3 | 3 | 64 | LReLU |
| conv2d | (64, 512, 64) | 3 | 3 | 64 | LReLU |
| downsample 2x2 | (32, 256, 64) | - | - | - | - |
| conv2d | (32, 256, 128) | 3 | 3 | 128 | LReLU |
| conv2d | (32, 256, 128) | 3 | 3 | 128 | LReLU |
| downsample 2x2 | (16, 128, 128) | - | - | - | - |
| conv2d | (16, 128, 256) | 3 | 3 | 256 | LReLU |
| conv2d | (16, 128, 256) | 3 | 3 | 256 | LReLU |
| downsample 2x2 | (8, 64, 256) | - | - | - | - |
| conv2d | (8, 64, 256) | 3 | 3 | 256 | LReLU |
| conv2d | (8, 64, 256) | 3 | 3 | 256 | LReLU |
| downsample 2x2 | (4, 32, 256) | - | - | - | - |
| conv2d | (4, 32, 256) | 3 | 3 | 256 | LReLU |
| conv2d | (4, 32, 256) | 3 | 3 | 256 | LReLU |
| downsample 2x2 | (2, 16, 256) | - | - | - | - |
| concat(x, minibatch std.) | (2, 16, 257) | - | - | - | - |
| conv2d | (2, 16, 256) | 3 | 3 | 256 | LReLU |
| conv2d | (2, 16, 256) | 3 | 3 | 256 | LReLU |
| pitch classifier | (1, 1, 61) | - | - | 61 | Softmax |
| discriminator output | (1, 1, 1) | - | - | 1 | - |

