# OpenReview forum: "GANSynth: Adversarial Neural Audio Synthesis"
_ICLR.cc/2019/Conference_

### Official Review · AnonReviewer1 · 2018-11-02
**Exciting work**

**Rating:** 8
**Confidence:** 3

**Review:**

This is an exciting paper with a simple idea for better representing audio data so that convolutional models such as generative adversarial networks can be applied. The authors demonstrate the reliability of their method on a large dataset of acoustic instruments and report human evaluation metrics. I expect their proposed method of preprocessing audio to become standard practice.

Why didn't you train a WaveNet on the high-resolution instantaneous frequency representations? In addition to conditioning on the notes, this seems like it would be the right fair comparison.

I'm still not clear on unrolled phase which is central to this work. If you can, spend more time explaining this in detail, maybe with examples / diagrams? In figure 1,  in unrolled phase, why is time in reverse?

Small comments:

- Figure 1 & 2: label the x-axis as time. Makes it a lot easier to understand.

- I appreciate the plethora of metrics. The inception score you propose is interesting. Very cool that number of statistically-different bins tracks human eval!

- sentence before sec 2.2, and other small grammatical mistakes. Reread every sentence carefully for grammar.

- Figure 5 is low-res. Please fix. All the other figures are beautiful - nice work!

---

> ### Author Response · Authors · 2018-11-16
> **Response to AnonReviewer1**
>
> Thank you for your time and insight in your review. We've incorporated changes to the paper and respond to your main points below:
>
> > “Why didn't you train a WaveNet on the high-resolution instantaneous frequency representations?”
>
> That’s an interesting avenue of research to explore. We trained WaveNets on the raw audio waveforms to provide strong and proven baseline models to compare against. Generating spectra with WaveNets is relatively unexplored and complicated by the high dimensionality at each timestep (number of frequencies * 2), each of which would have to be quantized in a traditional autoregressive treatment. It’s quite possible that 2 dimensional convolutions and autoregression could help overcome this, but then the model would most resemble pixelCNN and be far from a proven audio generation method for a strong baseline.
>
> > “I'm still not clear on unrolled phase which is central to this work. If you can, spend more time explaining this in detail, maybe with examples / diagrams? In figure 1,  in unrolled phase, why is time in reverse?”
>
> Apologies for the confusion. To help clarify, we’ve renamed the “unrolled” phase as “unwrapped” throughout the paper, which is better alignment to standards in the literature and popular software packages such as Matlab and Numpy (for example https://www.mathworks.com/help/dsp/ref/unwrap.html). We have also added text further describing figure 1 (2nd to last paragraph of introduction) to help explain unwrapping to be the process of adding 2*Pi to the wrapped phase whenever it crosses a phase discontinuity such as to recover the monotonically increasing phase. The time derivative of this unwrapped phase is then the radial instantaneous frequency.
>
> > “Figure 1 & 2: label the x-axis as time. Makes it a lot easier to understand.
>
> Thank you for the helpful pointer. We’ve added time axis labels to the figures and have also labeled the interpolation amounts for the interpolation figure.
>
> > “sentence before sec 2.2, and other small grammatical mistakes. Reread every sentence carefully for grammar.”
>
> We have read through the paper several times to revise grammatical mistakes including the sentence you highlighted.
>
> > “Figure 5 is low-res. Please fix. All the other figures are beautiful - nice work!”
>
> Thanks for catching this! We’ve updated the figure to be high resolution.

---

### Official Review · AnonReviewer2 · 2018-11-03
**Interesting take on GAN audio synthesis - accept**

**Rating:** 7
**Confidence:** 4

**Review:**

This paper proposes a strategy to generate audio samples from noise with GANs. The treatment is analogous to image generation with GANs, with the emphasis being the changes to the architecture and representation necessary to make it possible to generate convincing audio that contains an interpretable latent code and is much faster than an autoregressive Wavenet based model ("Neural Audio Synthesis of Musical Notes with WaveNet AutoEncoders" - Engel et al (2017)). Like the other two related works (WaveGAN - "Adversarial Audio Synthesis" - Donahue et al 2018) and the Wavenet model above, it uses the NSynth dataset for its experiments.

Much of the discussion is on the representation itself - in that, it is argued that using audio (WaveGAN) and log magnitude/phase spectrograms  (PhaseGAN) produce poorer results as compared with the version with the unrolled phase that they call 'IF' GANs, with high frequency resolution and log scaling to separate scales.

The architecture of the network is similar to the recently published paper  (Donahue et al 2018), with convolutions and transpose convolutions adapted for audio. However, there seem to be two important developments. The current paper uses progressive growing of GANs (the current state of the art for producing high resolution images), and pitch conditioning (Odena et al, where labels are used to help training dynamics).

For validation, the paper presents several metrics, with the recently proposed "NDB" metric figuring in the evaluations, which I think is interesting. The IF-Mel + high frequency resolution model seems to outperform the others in most of the evaluations, with good phase coherence and interpolation between latent codes.

My thoughts:
Overall, it seems that the paper's contributions are centered around the representation (with "IF-Mel" being the best). The architecture itself is not very different from commonly used DCGAN variants - the authors say that using PGGAN is desirable, but not critical, and the use of labels from Odena et al.

Many of my own experiments with GANs were plagued by instability (especially at higher resolution) and mode collapse problems without special treatment (largely documented, such as adding noise, adjusting learning rates and so forth). To this end, what do the authors see as 'high' resolution vis a vis audio signals?

I am curious if we can adapt these ideas for recurrent generators as might appear in TTS problems.

I rate this paper as an accept since this is one of the few existing works that demonstrate successful audio generation from noise using GANs, and  owing to its novelty in exploring representation for audio.

---

> ### Author Response · Authors · 2018-11-16
> **Response to AnonReviewer2**
>
> Thank you for your time and expertise in your review, we've addressed the key points below:
>
> > “...what do the authors see as 'high' resolution vis a vis audio signals?”
>
> In the context of these audio datasets, we use “high” resolution to refer more to the dimensionality of the signal to model with a single latent vector, rather than the temporal resolution of the audio. The spectral “images” that GANSynth models, have 1024 frequencies, 128 timesteps, and 2 channels, [1024, 128, 2], which is roughly equivalent to a [295, 295, 3] RGB image. This puts the task comparable to some of the higher-resolution GANs for images.
>
> > “I am curious if we can adapt these ideas for recurrent generators as might appear in TTS problems.“
>
> We agree that would be an interesting development. Recurrent generators, and even discriminators, would allow for variable-length sequences and variable-length conditioning as is common in speech synthesis or music generation beyond single notes. Our initial experiments at using recurring generators were not very successful, so we opted to adopt a better tested architecture for this study, but this is definitely still an area ripe for exploration.

---

### Official Review · AnonReviewer3 · 2018-11-06
**This paper proposes an approach that uses GAN framework to generate audio.**

**Rating:** 6
**Confidence:** 3

**Review:**

This paper proposes an approach that uses GAN framework to generate audio through modeling log magnitudes and instantaneous frequencies with sufficient frequency resolution in the spectral domain. Experiments on NSynth dataset show that it gives better results then WaveNet. The most successful deep generative models are WaveNET,  Parallel WaveNet and Tacotran that are applied to speech synthesis, the method should be tested for speech synthesis and compared with WaveNet, Parallel WaveNet as well as Tacotran.

For WaveNet, the inputs are text features, but for Tacotran, the inputs are mel-spectrogram. Here the inputs are log magnitudes and instantaneous frequencies. So the idea is not that much new.

GAN has been used in speech synthesis, see
Statistical Parametric Speech Synthesis Incorporating Generative Adversarial Networks
IEEE/ACM Transactions on Audio, Speech, and Language Processing ( Volume: 26 , Issue: 1 , Jan. 2018 )

So for this work, GAN's application to sound generation is not new.

---

> ### Author Response · Authors · 2018-11-16
> **Response to AnonReviewer3**
>
> Thank you for your review. We've done our best to address your concerns with paper revisions and in the comments below:
>
> > “The method should be tested for speech synthesis and compared with WaveNet, Parallel WaveNet as well as Tacotran”
>
> We agree that it would be very interesting to adapt these methods to speech synthesis tasks, but believe that this lies outside the scope of this initial paper on adversarial audio synthesis. As we note in AnonReviewer2’s comments, adapting the current methods to incorporate variable-length conditioning and generate variable-length sequences is a non-trivial extension and requires further research. In the context of this study, we’ve done our best to provide strong autoregressive baselines from state-of-the-art implementations of WaveNet models (including 8-bit and 16-bit output representations).
>
> Thank you for highlighting that this is an important direction for this research. We have updated the text of the paper with a paragraph highlighting the importance and difficulty of pushing the current methods forward for more general audio synthesis tasks.

---

### Author Response · Authors · 2018-11-16
**Updates**

We would like to thank all the reviewers for their thoughtful and helpful reviews. In addition to answering the points of each individual reviewer below, we also want to highlight several additions we have made to the appendix to hopefully improve clarity and reproducibility.

* An additional figure displaying spectrograms for a Bach Prelude synthesized both with and without latent interpolation, the audio for which can be found in the supplemental.
* Substantial experimental details to improve reproducibility, including detailed architecture parameters and training procedures.
* An additional NDB figure highlighting the lack of diversity of WaveNet baseline samples.
* A table of additional baseline comparisons, justifying the use of WaveGAN and 8-bit WaveNet as the strongest baselines.

---

### Meta-Review · Area_Chair1 · 2018-12-13
**novel approach with good results shown by extensive evaluation**

**Confidence:** 4
**Recommendation:** Accept (Poster)

**Metareview:**

1. Describe the strengths of the paper.  As pointed out by the reviewers and based on your expert opinion.

- novel approach to audio synthesis
- strong qualitative and quantitative results
- extensive evaluation

2. Describe the weaknesses of the paper. As pointed out by the reviewers and based on your expert opinion. Be sure to indicate which weaknesses are seen as salient for the decision (i.e., potential critical flaws), as opposed to weaknesses that the authors can likely fix in a revision.

- small grammatical issues (mostly resolved in the revision).

3. Discuss any major points of contention. As raised by the authors or reviewers in the discussion, and how these might have influenced the decision. If the authors provide a rebuttal to a potential reviewer concern, it’s a good idea to acknowledge this and note whether it influenced the final decision or not. This makes sure that author responses are addressed adequately.

No major points of contention.

4. If consensus was reached, say so. Otherwise, explain what the source of reviewer disagreement was and why the decision on the paper aligns with one set of reviewers or another.

The reviewers reached a consensus that the paper should be accepted.